behaviour, ecology, environmental science

social networks, demography, natural disasters, mortality, dynamic communities, predation

**Author for correspondence:**
Julian C. Evans
e-mail: jevansbio@gmail.com

# A natural catastrophic turnover event: individual sociality matters despite community resilience in wild house mice

Julian C. Evans[1], Jonas I. Liechti[2], Bruce Boatman[1] and Barbara König[1]

[1]Department of Evolutionary Biology and Environmental Studies, University of Zurich, Winterthurerstrasse 190, 8057 Zurich, Switzerland
[2]Institute for Integrative Biology, ETH Zurich, Universitätsstrasse 16, 8092 Zurich, Switzerland

JCE, 0000-0001-6810-199X; JIL, 0000-0003-3447-3060; BK, 0000-0001-7696-4736

Natural disasters can cause rapid demographic changes that disturb the social structure of a population as individuals may lose connections. These changes also have indirect effects as survivors alter their within-group connections or move between groups. As group membership and network position may influence individual fitness, indirect effects may affect how individuals and populations recover from catastrophic events. Here we study changes in the social structure after a large predation event in a population of wild house mice (*Mus musculus domesticus*), when a third of adults were lost. Using social network analysis, we examine how heterogeneity in sociality results in varied responses to losing connections. We then investigate how these differences influence the overall network structure. An individual's reaction to losing associates depended on its sociality prior to the event. Those that were less social before formed more weak connections afterwards, while more social individuals reduced the number of survivors they associated with. Otherwise, the number and size of social groups were highly robust. This indicates that social preferences can drive how individuals adjust their social behaviour after catastrophic turnover events, despite the population's resilience in social structure.

## 1. Introduction

Species in the wild are subject to sudden and unpredictable events that cause rapid and large-scale changes to population demography. 'Catastrophic turnover events' [1] such as disease epidemics or natural disasters can cause an unusually high degree of mortality or dispersal over a short time frame [2]. This has the potential to dramatically impact group social structure [3]. In social animals, the stability of associations will vary between both individuals and species. Some species may possess extremely resilient social associations that remain consistent over time [4,5], while others might change between seasons or contexts [6–8]. Therefore, while in some species the structure of social groups may be more robust to changes in demography [9], in others associations can change entirely when key individuals are removed [3,10]. Such changes in social structure can both impact individual fitness and result in knock-on effects that impact the population as a whole [11]. Thus, understanding the resilience of social structures over time, contexts and in response to disturbances and ecological change, is crucial for our understanding of the social processes driving these associations and their influence on life history [1].

Social network analysis provides an excellent framework to examine these types of events, allowing the changes in social structure to be described via the change in individual edges between nodes (pairwise connections between two individuals). After a catastrophic turnover event, those that remain will often suffer the direct effect of having a reduced number of associates, as a result of

losing connections with those that have left or died. However, individuals can also be affected by the indirect effects of a turnover event, as survivors adjusting their social associations with each other leads to changes in social structure [1,12]. For example, some individuals may alter their behaviour to seek out new associates to replace those they lost, or exploit a gap in the network to alter their social standing, leading to the merging of social groups [1,13,14]. Alternatively, some individuals might become more socially insular, interacting only with remaining strong associates, leading to greater fragmentation of a network [1,3]. These changes may be more likely if the individual lost occupied a highly central position in a group or acted as a 'bridge' between social groups [1,12,15]. Exactly how a social network changes will therefore depend on the species' life history and social behaviour, the identity of the individuals that are lost and on individual social preferences. Depending on the resilience of a network, social structure may return to something similar to that before the disaster. Alternatively, the new network that emerges could result in individuals experiencing altered selection pressures, dependent on the fitness benefits of sociality in that species [1,10,16]. Knowledge of how social structure will change in response to catastrophic turnover events would thus be crucial when considering how different types of individuals in a population might be affected by ecological changes or disasters.

It is, however, difficult to collect data on the impact of such catastrophic events on network structure. Unlike disease, where mortality typically extends over a longer period of time, natural disasters generally occur over far shorter time periods. Much of the current empirical knowledge about how a demographic turnover event can alter network structure comes from removal experiments [6,17,18]. While these results have been extremely informative and have the advantage of being able to be replicated multiple times and target specific individuals, carrying out large-scale removals over a short period of time to simulate a disaster is ethically and logistically challenging. Additionally, comparisons between experimental and natural removal of individuals have suggested some discrepancies [1,18,19]. Being in a position to record data over a sufficient time period before and after a disaster is rare, generally only being possible in long-term studies where data are collected continuously [3,16,20].

Here we examine changes in social structure after a sudden, large-scale cat predation event in a long-term field study of a population of wild house mice (*Mus musculus domesticus*). Based on previous empirical studies we assumed that groups that suffered a higher loss of individuals would be more likely to fragment, possibly leading to a growth in size of less affected groups [21]. We also predicted that individuals that were more sociable prior to the event would attempt to replace their lost connections over time, by initially moving between groups, leading to increased betweenness centrality. Finally, we predicted that individuals would eventually return to associating with a similar number of individuals as before, occupying a network position close to the one they occupied pre-event.

## 2. Material and methods

### (a) Study system

Our study system is a population of wild house mice living in a barn near Zurich, Switzerland. This population has been intensely studied for over 15 years and currently numbers approximately 700 mice (adults, subadults and pups). The set-up within the barn mimics the situation of wild house mice that live commensally with humans in stables, barns or houses in middle Europe, with access to food and various shelters. The barn consists of a single large 72 $m^2$ space divided into four sections by low barriers (see electronic supplementary material, figure S1). Mice can leave and enter the building under the roof and through gaps in the wall; they can also freely move between sections through holes in the barriers or by climbing over them. Each section contains 10 artificial nest-boxes fitted with radio frequency identification antennae. We attempt to equip all adults (weighing ≥ 18 g) with passive integrated transponder tags during regular population checks, every six to eight weeks. For more details on data collection in the population and on the antennae system, see König, *et al.* [22]. Food, water and nesting materials are provided and nest-boxes are regularly monitored for litters. Other than this, the population is left to develop naturally, with individuals free to move about the barn or disperse. Mice that belong to the same social group meet in spatially clustered nest-boxes (between one and eight), generally located within one section. However, individuals are regularly observed also using nest-boxes in neighbouring sections or permanently moving between sections [23]. The number of mice sharing a single nest-box varies throughout the year, with larger groups observed during the off-breeding season in winter than during the breeding season in summer (up to 28 tagged adults per nest-box [23,24]).

### (b) Predation event

Though we attempt to prevent any animals larger than a mouse from entering the barn, during the weekend of 19–20 January 2019 at least one cat managed to enter the building. One hundred and sixteen mice were found dead on 21 January, of which 85 were tagged. An additional 107 individuals also subsequently disappeared from the antennae system recordings. These missing individuals either never had their bodies recovered (despite intensive searches) or possibly dispersed from the barn owing to the perturbation. Out of the 478 tagged individuals recorded as present before the event, the total number of missing or dead tagged adults (henceforth just 'missing') was 192. No dead pups were found as attack took place during the off-breeding season, with no litters having been found for the previous seven weeks.

### (c) Network construction

Networks were constructed using simple ratio indexes [25], based on the proportion of time (in seconds) two individuals spent in a nest-box together. Each network consisted of 5 days of data. A full network constructed using data from the 5-day window prior to the attack was used to ascertain individuals' network positions immediately prior to the predation event (full pre-event network, figure 1a). A further four networks were generated representing the timesteps following the predation event. These networks consisted only of individuals that appeared in all timesteps and were therefore not missing after the predation event (hereafter 'survivors'). Five-day windows were chosen so as to have sufficient data to be confident as to the strength of associations while still detecting any short-term changes caused by the predation event. All antennae data recorded during the weekend of the predation event were excluded. A version of the pre-event network restricted only to these surviving individuals was also created. The main networks used for analysis therefore consisted of six survivor-only networks of 243 individuals (two from before the event and four after, figure 2a–f) and one full pre-event network of 478 individuals (figure 1a). A further full pre-event network from two timesteps prior to the attack was used during dynamic community detection. All networks were constructed in R [26] using the igraph package [27]. For each individual, we

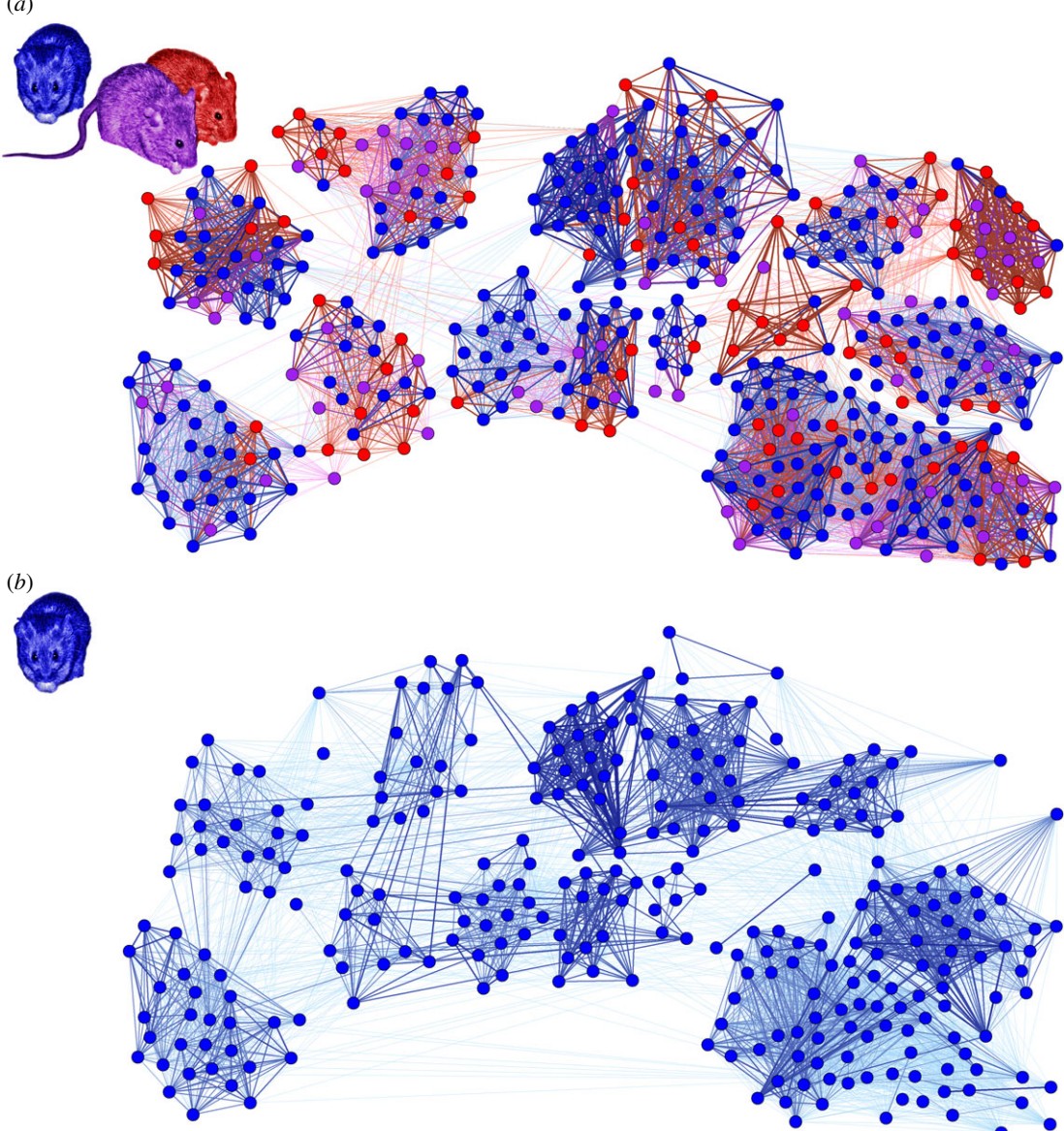

**Figure 1.** Full network in the timestep (*a*) 5–1 days immediately before the predation event (timestep 2), and (*b*) 1–5 days immediately after the predation event (timestep 3). Thicker/darker lines (edges) denote stronger social associations. Red nodes are individuals found dead after the event (*n* = 85), purple nodes individuals that went missing (*n* = 107) and blue nodes survivors (*n* = 286). Red edges are connections to dead individuals, purple edges connections to missing individuals and blue edges connections to survivors. Node position is approximately based on individuals' nest-box use prior to the predation event.

calculated the proportion of its associations made up of missing individuals in the full pre-event network by summing the total strength of edges to missing individuals and dividing it by their total edge strength (figure 1). Each individual in each network had its degree, weighted degree and betweenness centrality calculated using igraph. Degree centrality represents the number of other mice an individual associated with in a timestep. This metric was used to check if survivors changed their general association patterns, meeting a greater number of individuals as they sought out new social connections or reducing their connections with other survivors if they became more insular. Weighted degree centrality is similar but weighted by the strength of these relationships, especially when combined with degree. A high degree centrality combined with a low weighted degree centrality would indicate an individual that met a large number of other individuals, but did not spend a large amount of time with any of them (as we might expect if an individual was prospecting for a new social group). An individual's betweenness centrality indicates the number of shortest pathways between all dyads in the network that pass through that individual. This value can often be used to distinguish individuals that associate with multiple social groups. We expected this value to temporarily increase in the survivor-only

networks for individuals that were seeking a new social group to join, while expecting it to decrease if social groups became more insular. By default, igraph inversely weights networks when measuring betweenness. We accounted for this by using pre-inverted versions of the network weights when calculating this network statistic. For each of the four survivor-only post-event networks (figure 2*c*–*f*) and the first survivor-only pre-event network (figure 2*a*), we calculated the difference in all network statistics from their values in the survivor-only network immediately prior to the event (figure 2*b*).

A thousand randomized versions of the post-event networks and the timestep 1 network (10–6 days before the event) were also created to compare any observed changes in network structure with that expected from individuals simply associating randomly. Individuals could potentially associate more randomly if they are behaving erratically after the event owing to stress, or interacting with others simply as a result of avoiding areas now perceived as dangerous [28–30]. Randomized networks were created by carrying out 11 000 node swaps (10 per randomized network), with the initial 1000 swaps used as a burn-in. Node swaps swap the identities of two individuals in the network, randomizing their social connections while preserving their attributes,

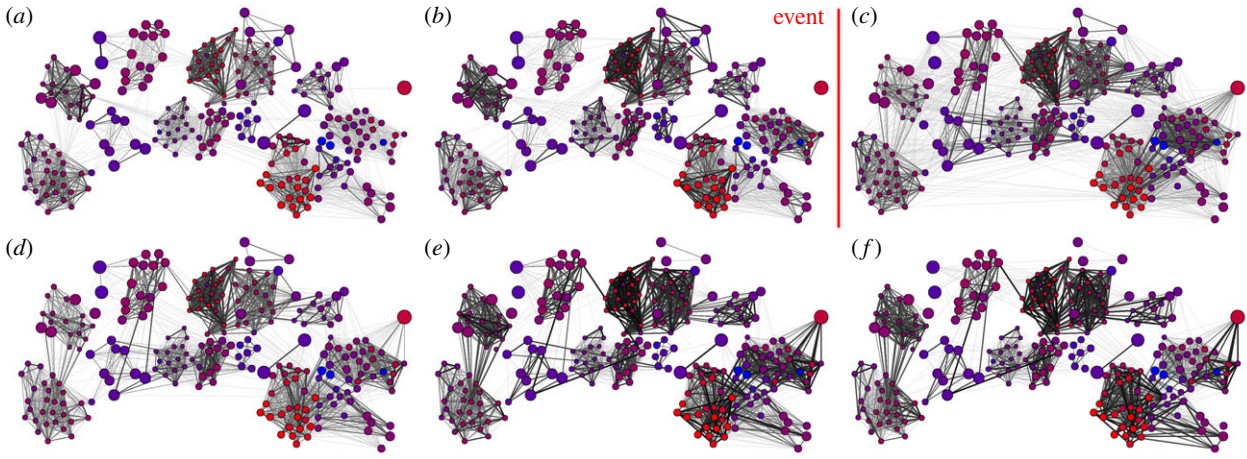

**Figure 2.** Within survivor networks (*a*) 2 timesteps before predation event, (*b*) 1 timestep before predation event, (*c*) 1 timestep after predation event, (*d*) 2 timesteps after predation event, (*e*) 3 timesteps after predation event and (*f*) 4 timesteps after predation event. Thickness of edges denotes strength of social connection. Colour of node represents pre-event weighted degree as calculated from the full network (figure 1*a*), where blue is a low pre-event weighted degree and red is a high pre-event weighted degree. Size of node indicates the proportion of association lost due to individuals going missing. Node position is approximately based on individuals' nest-box use prior to the predation event.

such as their pre-event sociality or connections lost [31]. In each of the randomizations, the same swaps were carried out in every network, so a pair of individuals swapped in one timestep would be swapped in all timesteps.

### (d) Dynamic community detection

In order to quantify the effects of the predation event on overall group structure, we used dynamic community detection to link detected communities between timesteps. This allowed us to measure changes in the size of social groups and how individuals moved between them over time. Within each network, individuals were organized into social groups using the clustering algorithm developed by Rosvall & Bergstrom [32] (electronic supplementary material, figures S2 and S3). The sequence of pre- and post-predation event networks was combined into a time-window graphs representation [33] of the population structure's temporal course. The method by Liechti & Bonhoeffer [34] was used to determine the temporal course of the social communities throughout this sequence. This method allows social groups to be mapped between timesteps based on reciprocal majority identification. We used networks from two timesteps prior to the predation event to identify these majorities. Membership of social groups and the resulting dynamic communities were then determined on all full networks (figure 3) and applied to the surviving individuals (electronic supplementary material, figures S3 and S4). We also considered the movement events between dynamic communities, i.e. the number of times one or more individuals changed their dynamic community associations between timesteps.

### (e) Analysis

Bayesian distributional regression models were fitted in R using the package brms [35]. For models of changes to individual network statistics, we fitted a three-way interaction between post-event timestep (fitted as a categorical variable), an individual's sociality (as measured by its weighted degree in the full pre-event network, figure 1*a*) and the proportion of total edge strength in the full network lost due to individuals dying or going missing (see figure 2 for a visualization). Individual ID was fitted as a random effect and all models were fitted with flat priors. The changes in degree, weighted degree or betweenness centrality were used as response variables. Additionally, we allowed the variance parameter of the response variable to change in relation to the interaction between pre-event sociality and proportion of pre-event edge strengths made up of missing individuals [36]. This allowed a more

parsimonious fit to the large amount of post-event variation in individual responses. All numeric explanatory variables were mean-centred and rescaled so that 1 was equal to 1 s.d. of the original variable. The same models were also fitted on the 1000 random versions of the post-event and timestep 1 networks. Effect sizes were then compared between the randomized networks and the real network. *P*-values were calculated as the proportion of models in which posterior effect sizes differed from the randomized networks [31]. In order to better understand whether there was an overall change in all individuals' network statistics in a particular timestep, as opposed to different types of individuals reacting differently at different timesteps (as represented by the interaction fitted in the main models which allows slopes to vary per timestep), we also fitted a version of each model where timestep did not interact with the other variables. Finally, though we believed that our measure of pre-event sociality would also reflect differences in sociality between the sexes, we examined whether males and females differed in their reaction by refitting the models of change in degree and weighted degree with sex replacing an individual's pre-event sociality. Non-convergence issues prevented the fitting of a model of change in betweenness centrality in relation to sex.

For community structure, we fitted a similar set of models. In this case, the explanatory variables were a three-way interaction between post-event timestep, the pre-event community size (as calculated from the full network) and the proportion of that community missing. The response variables for these models were the change in community size (as calculated from the survivor-only networks) and within-community edge density (as calculated from the survivor-only networks). This measured the level of connectedness within the group, as a proportion of all possible connections, allowing us to quantify the extent to which members of the group were associating with other members of the group. We expected this value to increase in networks in which more individuals were lost, as individuals increased their level of connectivity with other survivors within the group. Group ID was fitted as a random effect.

## 3. Results

### (a) Overall network structure

After the predation event 192 individuals were missing that had been recorded by the antennae system before the event (85 tagged individuals found dead, 107 disappeared). The number of individuals recorded was thus reduced from 478 to

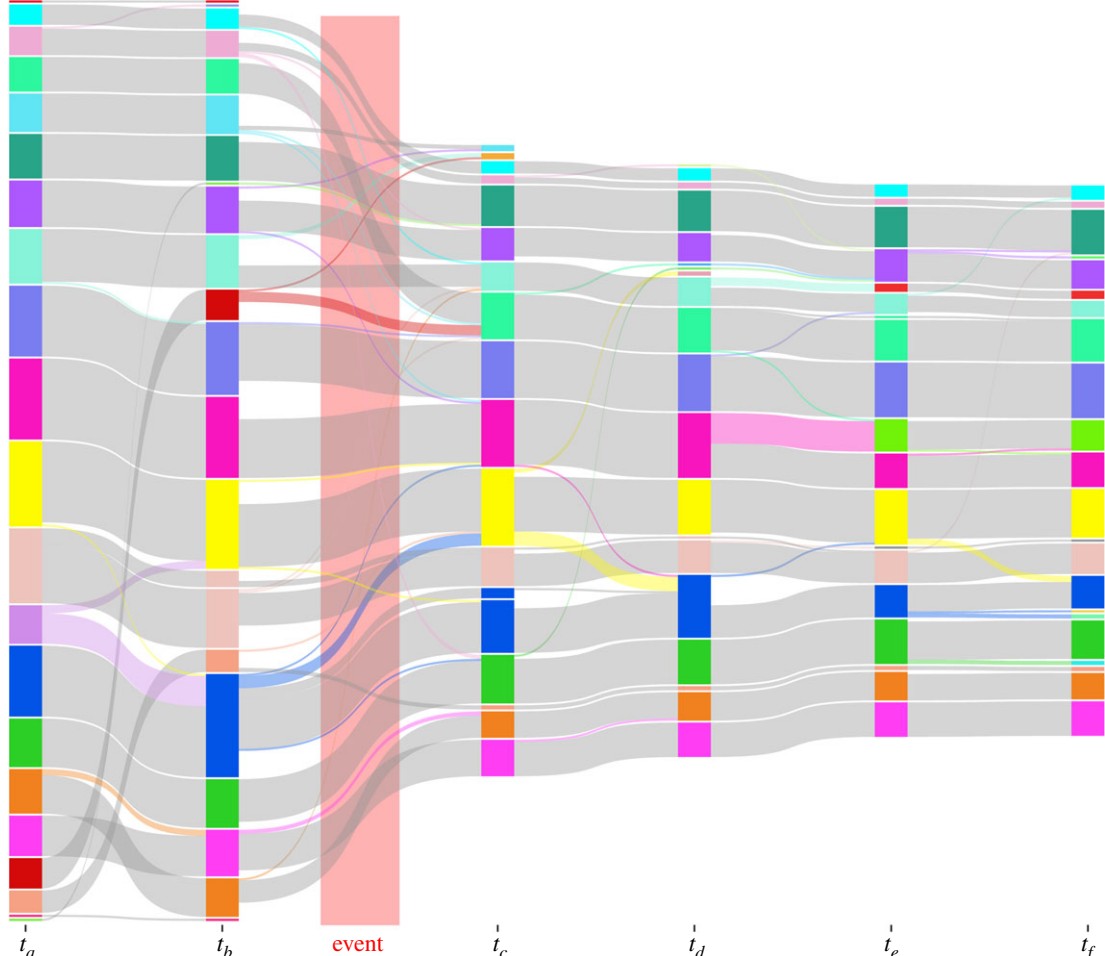

**Figure 3.** Alluvial plot illustrating the movements between dynamic communities present in the full networks at (*a*) 2 timesteps before predation event, (*b*) 1 timestep before predation event, (*c*) 1 timestep after predation event, (*d*) 2 timesteps after predation event, (*e*) 3 timesteps after predation event and (*f*) 4 timesteps after predation event. Movements from one community to another are highlighted in the same colour as the source community. Figure represents full unrestricted networks (see electronic supplementary material, figure S2 for network diagram; see electronic supplementary material, figure S4 for an illustration based on the within survivor networks). (Online version in colour.)

286 (figure 1). Of the recorded individuals lost, 94 were female and 98 were male, leaving 157 females and 129 males. Of these, 136 females and 107 males were present in sufficient time-steps to be included in analysis. Edge density increased from 0.07 prior to the event to 0.11 after the event. Edge density followed a similar pattern in survivor-only networks, increasing from 0.09 (pre-event) to 0.12 (post-event). However, despite this, the number of social groups in the network remained relatively stable. Prior to the predation event, the full network had 17 groups consisting of three or more individuals, while in the net-work immediately after the attack 14 groups remained (figure 3). All communities were affected by the event, each having at least one individual found dead or gone missing (electronic sup-plementary material, figure S2). The number of individuals lost per community ranged from two individuals, correspond-ing to only 12% of the community, up to 21 individuals in the largest community, corresponding to 42%. The biggest relative impact was in a community consisting of 17 individuals before the attack that was reduced to only two individuals, a loss of 89.5% (electronic supplementary material, figure S2).

## (b) Individual sociality

Here we present the results of the model of changes in individ-ual sociality, both with (electronic supplementary material, tables S1–S3) and without (electronic supplementary material,

tables S4–S6) the three-way interaction. The model of simple degree centrality suggested that most individuals increased their degree slightly in the timestep immediately following the attack, in particular if individuals had lost more of their social connections (electronic supplementary material, table S1; estimated positive change in degree in figure 4*a*). However, we cannot exclude that this increase in degree was due to random meetings, as the effect size did not differ significantly from randomizations (electronic supplementary material, tables S1 and S4). Over time, those that did not lose many con-nections appeared to return to a degree centrality similar to what they had pre-event (individuals on the left-hand side of each panel, figure 4*a*). However, lower sociality individuals that had lost more associates continued to show an increase in the number of individuals they interacted with over the post-event period studied (change in slopes over time, figure 4*a*). At the same time, more social individuals that lost more connec-tions showed a decrease in their degree over time, particularly in the last two timesteps examined. The model where slopes were not fitted per timestep indicated a general decrease in degree centrality among all surviving individuals in the last two timesteps (electronic supplementary material, table S4). The variance estimates suggested that variation in the response to losing connections increased among those that had been more social before the attack (electronic supplemen-tary material, tables S1 and S4).

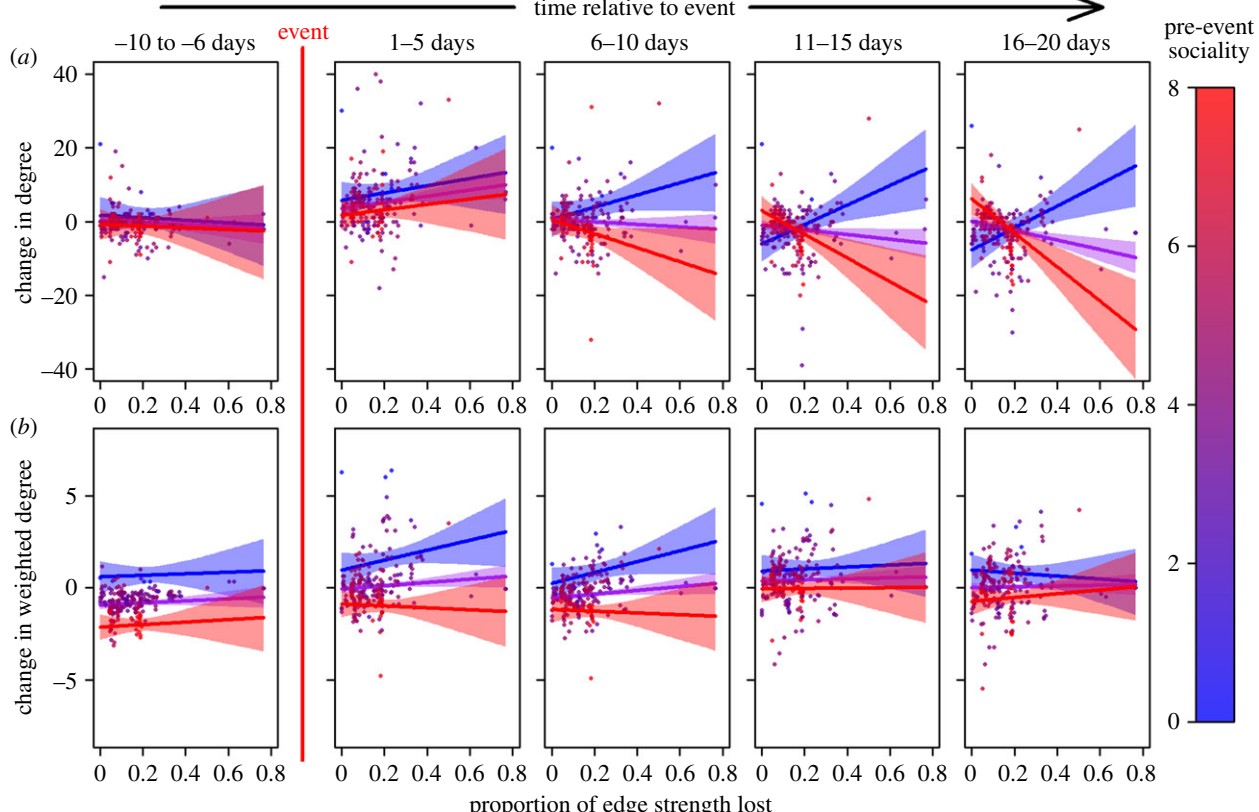

**Figure 4.** Individuals' change in (a) degree centrality and (b) weighted degree centrality in the timestep immediately before the predation event (timestep 2, 5–1 days before predation event, is not pictured owing to its use as a point of comparison) within the survivor-only network in relation to the edge strength lost via associates going missing and their pre-event sociality (weighted degree centrality as calculated from the full network, indicated by colour) and timesteps relative to the predation event, with model estimates for minimum (blue), mean (purple) and maximum (red) pre-event sociality and 95% confidence intervals.

In the model of weighted degree centrality, individuals of lower pre-event sociality initially appeared to be more likely to slightly increase their weighted degree immediately after the attack if they lost a higher proportion of associates (figure 4b). Conversely, those that had been more social pre-event and lost a higher number of associates were estimated to somewhat reduce their weighted degree. Over time, however, the strength of these effects seemed to the decrease and individuals started to return approximately to the weighted degree immediately prior to the predation event (electronic supplementary material, table S2; change in slopes over time in figure 4b). Though the effect of actually losing associates was stronger than expected from random associations it was a relatively weak effect, suggesting that even those that did not directly lose connections altered associations with other survivors. As with degree, variance estimates predicted that individuals that had lost more connections would vary more in their response to the predation event (electronic supplementary material, table S2). Neither an individual's proportion of edge strength lost nor its pre-event sociality was a clear predictor for changes in its betweenness centrality (electronic supplementary material, table S3). The models did suggest, however, that variation in changes in betweenness increased depending on the proportion of connections that had been lost, regardless of sociality (electronic supplementary material, table S3). Results from the models replacing pre-event sociality with sex did not find any clear differences between how males and females altered their degree centrality (electronic supplementary material, table S7), but did suggest that in later timesteps males were more likely to have a reduced weighted degree than females (electronic supplementary material, table S8).

## (c) Community structure

Comparing the group structure in the networks immediately before and after the attack revealed a reduction in the modularity [37] and thus a decrease in the separation of the population into social groups. This pattern was apparent in both the full networks (modularity before: 0.613, after: 0.534; see electronic supplementary material, figure S5) and the survivor networks (modularity before: 0.674, after: 0.583; see electronic supplementary material, figure S5). This decrease in modularity was present only in the timestep immediately following the attack.

Immediately after the event, 261 out of 286 surviving individuals remained in the same dynamic communities as before the event. The number of movement events between dynamic communities was computed for the survivor-only networks: between the two pre-event networks, between the networks immediately before and after the attack, and between the four networks after the attack (see electronic supplementary material, figure S4). In the time period prior to the event, only three movements of individuals among communities were registered between the two networks, none of which was between groups in different sections. As expected, the number of movements between the pre- and post-attack network was highest, with 16 movements, ranging from 1 to 6 individuals moving to new communities. Five of these movements were between groups in different barn sections. Subsequently, 5 (1 between sections), 8 (1 between sections) and 10 (2 between sections) movements between dynamic communities occurred between the three consecutive post-event networks.

Neither the proportion of individuals a group lost, the group size in the full pre-event network prior to the attack nor the interaction between these terms was found to be a strong predictor for changes in group size or within-group edge density in the survivor-only networks (electronic supplementary material, tables S4 and S5). Models using the proportion of killed individuals instead of proportion missing produced the same result (electronic supplementary material, tables S9 and S10).

## 4. Discussion

A sudden, unpredictable and catastrophic predation event had pronounced effect on the social structure of a population of wild house mice. Our study allowed us to use detailed social association data to measure changes in social networks after the loss of approximately a third of the adult population, and to examine how these changes evolved over time. This dataset presents a rare opportunity to study the aftermath of a natural catastrophic turnover event. Individuals both increased and decreased their level of social association with other survivors over time, demonstrating the longer-term indirect effects of loss of associates. Interestingly, how individuals changed their social connections was based on their pre-event level of sociality. Individuals increased their number of weak connections if they had previously shown low sociability, but strengthened ties with a reduced number of associates if they had previously been highly social. However, despite this, the structure of remaining communities stayed stable.

The observed changes in individual degree and weighted degree in our pre- and post-event survivor networks indicate how individuals formed new connections in the wake of the predation event. With increasing time after the attack, mice with lower pre-event sociality increased their number of associates (degree) but not the overall strength of those associations (weighted degree) when they had lost a high proportion of associates. This suggests the formation of a large number of weak connections with other survivors. These individuals may already have been on the periphery of groups that were mostly destroyed or that fractured owing to the removal of key individuals [1,10], resulting in them joining a surviving social group and beginning to associate with a new set of individuals. Individuals might particularly be driven to join a new social group if they had lost preferred associates [38]. Individuals joining new social groups, particularly those based in different sections of the barn, may also have abandoned preferred nest-boxes owing to them now being perceived as unsafe. However, we assume social rather than nest-box preferences are the most important drivers of group membership. Liechti et al. [23] observed that social groups often persist longer than the lifespan of their members. Membership of a specific group that can successfully defend a territory containing several nest-boxes might be more important than choosing a specific nest-box. Alternatively, the increase in the number of weak connections for less social individuals may be the result of an influx of members from another group that was mostly destroyed. In both scenarios, these individuals maintain a similarly low level of sociality with survivors as prior to the attack, possibly owing to their status or personal preferences [39,40]. Conversely, individuals with higher pre-event sociality appeared to do the opposite. They reduced their number of connections with other survivors over time (degree), while

returning to a similar level of sociality (weighted degree). This could be indicative of these individuals becoming more socially insular, spending more time in nest-boxes with a smaller number of remaining associates. Similar results have been found in other species, with individuals increasing the strength of their connections after a disaster [3,20] or experimental removal [13]. Additionally, among less social individuals a proportion of edge strength lost might also represent losing a large number of their associates, as opposed to a more social individual which might lose a similar proportion of edge strength but have a greater number of associates remaining. This might lead to less social individuals becoming more mobile immediately after the attack, owing to greater disruption to their social structure, leading to them encountering more individuals and increasing their degree in a random manner. When examining if the response to the event differed between sexes, males appeared to differ in the final timesteps, being more likely to have a lower weighted degree. This may be due to males being generally less social than females or being unable to form new connections as quickly in a new social group owing to competition with other males [41].

We expected to see an increase in individuals' betweenness centrality within the post-event survivor network as individuals looked for a new social group to join. Although there appeared to be an increase in the number of edges between groups from different sections of the barn, we did not find any clear effect, similar to a removal experiment in a large bird population which had also reported lack of change in betweenness [13]. How individuals changed in their betweenness could not be estimated by their prior sociality or how affected they were by the predation event. Nevertheless, some individuals altered their betweenness in our study. Those that had lost more connections showed greater variation in how their betweenness changed. It is possible that any increase in betweenness observed might instead best be predicted by some individual trait such as exploratory personality [42], social phenotype [39,40] or prior social status [43].

Despite the event affecting all social clusters, groups did not collapse, as might be expected if individuals began consistently associating more randomly for a significant length of time after the perturbation owing to stress imposed by the threat of predation, becoming less choosy with whom they associated (in order to gain dilution benefits) or using less space within the barn owing to certain areas being perceived as dangerous [1,30]. Within surviving individuals, community structure remained stable throughout the study period. The changes in size or density of connections within a cluster seemed unrelated to the loss of individuals within a group. We did observe both a reduction in the modularity of the social structure and an increase in between-community movements after the event. Taken together these results may suggest that, while the social clusters did not collapse, they seemed to temporarily become less well defined and more permeable over the entire post-attack observation range. This might indicate that group structure and therefore overall network structure may be far more resilient than assumed. This would be beneficial to many group members given the potential influence of social position on reproduction [43,44], general survival [45,46] and survival of further catastrophic turnover events [16,45]. Stable social groups in the wake of such an event would also mean that any benefits conveyed by grouping behaviour such as predator defence [47,48] or thermoregulation [49] will be maintained despite a loss of a substantial part of the population.

It is uncertain to what extent the impact of a large predation event on network structure might differ from other catastrophic turnover events. As already mentioned, obtaining natural data on rapid demographic change is extremely rare. Previous studies have looked at network structure in the aftermath of fires [20] and hurricanes [3]. Though predation is an extremely common event, network studies of its impact are comparatively rare. Carter *et al.* [38] looked at kangaroo (*Macropus giganteus*) networks after members were predated, suggesting that individuals might be less selective in their associations after losing group members. Similarly a study of chacma baboons (*Papio hamadryas ursinus*) also suggested that individuals would attempt to interact with a greater number of individuals after losing close associates to predation [50]. Removal experiments will be needed to analyse whether the changes we observed in our study population were specifically due to the loss of individuals or due to the disturbance and stress caused by a significant predation event. Furthermore, as with many network studies, our networks only represent interactions in a single context (sharing nests) at fixed locations. Future study could examine how a turnover event influences individuals in different contexts such as foraging or the frequency of aggressive interactions [51,52]. This also raises the question of how changes in the social structure of species that range over larger areas and interact in a variety of different locations might differ from our observed results [1].

In conclusion, this study expands our knowledge of how animal social structure will be altered by catastrophic turnover events. Our results emphasize the impact of the direct loss of associates but also illustrate the indirect effects this will have on the social networks of those that survive. Our findings suggest that individuals' social preferences result in differences in how they react in the aftermath of such an event, and how quickly overall group structure might stabilize. This has important implications for how such perturbations will affect surviving individuals' fitness. It is particularly important given the increasing number of species facing rapid ecological changes [53]. Further work to look at the longer-term effects, particularly continued survival and reproductive success in the following breeding season, will help to better understand the impact of rapid demographic change.

Ethics. The barn is run in accordance with Veterinary Office Zurich guidelines and subject to the Swiss animal protection law (permit no. ZH091/16 from Kantonales Veterinäramt Zürich).

Data accessibility. Data are available as part of the electronic supplementary material.

Authors' contributions. The study was conceived by J.C.E. Fieldwork was carried out by B.B., with some assistance from B.K., J.C.E. and colleagues. Dynamic community analysis was carried out by J.I.L., with other analyses performed by J.C.E. The paper was written by J.C.E. and B.K., with all authors contributing to revisions.

Competing interests. We declare we have no competing interests.

Funding. Financial support was provided by the Swiss National Science Foundation, grant no. 31003A_176114, and the University of Zurich.

Acknowledgements. We thank all co-workers and helpers who contribute to running the barn project and contribute to data collection. We also thank Jordi Bascompte as well as two reviewers for helpful comments on the manuscript.

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
