## [Reviewer comments · Proceedings of the Royal Society B: Biological Sciences]

Review History

RSPB-2019-2880.R0 (Original submission)

Review form: Reviewer 1

Recommendation

Major revision is needed (please make suggestions in comments)

Scientific importance: Is the manuscript an original and important contribution to its field?

Excellent

General interest: Is the paper of sufficient general interest?

Excellent

Quality of the paper: Is the overall quality of the paper suitable?

Excellent

Is the length of the paper justified?

Yes

Should the paper be seen by a specialist statistical reviewer?

No

Do you have any concerns about statistical analyses in this paper? If so, please specify them explicitly in your report.

No

It is a condition of publication that authors make their supporting data, code and materials available - either as supplementary material or hosted in an external repository. Please rate, if applicable, the supporting data on the following criteria.

Is it accessible?

N/A

Is it clear?

N/A

Is it adequate?

N/A

Do you have any ethical concerns with this paper?

No

Comments to the Author

This study leverages a catastrophic predation event in a well-studied wild population of house mice to explore the consequences of sudden, massive population decline on social network structure. Fortunately, the social structure of the population had been comprehensively monitored using a continuous RFID data logging system as part of a long-term study prior to the predation event. Since the monitoring could continue with the surviving mice, the researchers could analyze in detail the stability of the social structure despite the loss of about 40% of the population. They also find that an individual's pattern of sociality prior to the predation event affects how their patterns of connectivity change after the event. This is certainly an interesting case study for linking demographic events to social structure in natural populations.

One inevitable limitation of this study, as with any study relying solely on RFID systems, is that the assessment of social structure is necessarily tied to specific locations seeded with key resources. Thus, the stability of the social network may be purely driven by the stability of an individual's preference for the nest boxes that it knows. This would be meaningful in its own way, so it doesn't necessarily take away from the study. However, the spatial aspect of network stability should be acknowledged throughout this paper. In the Discussion, it may be worth also considering how response to catastrophic mortality events in location-based social networks like this one may differ from other systems in which social associations can occur over broader spatial areas.

More minor comments:

Lines 45-49: The two sentences here sound redundant with each other. Perhaps there needs to be more explanation of what "indirect effects" are, and how they are distinct from loss of connections due to death.

Line 94: The spatial structure of the barn and nest box array should be described in the paper. If there is no room, it could be presented as a supplement. Without this detail, it makes the overall patterns more challenging to interpret. For example, it is stated here that the mice are free to move and disperse. Upon reading the study that describes the methodology in full (König et al. 2015), it seems that the setup is slightly more complex. For example, the barn is segmented into 4 parts, partitioned by walls that have holes that mice can move through. While this may not limit dispersal per se, it likely restricts movement patterns significantly, which would likely change how individuals use nest boxes. I'm not saying that this is necessarily a major flaw in the study – any natural system will also have structures that affect movement patterns, and thus

social structure. My main point is that this spatial arrangement should be described within this paper in order to make it easier for the readers to interpret the results.

Line 190: I did not see Table 1 in the manuscript.

Line 190-194; Lines 200-202: It took me a while to get that these sentences are referring to the changes in the slope or intercept in panels in Figure 4 going left to right on each row. I think there needs to be much more explanation either in the main text or in the figure captions.

Line 221: "Unsurprisingly" □ "As predicted"?

Figure 4: I could not really interpret this figure because the color scales were not explained. The color gradient on the right seems to correspond to the color of points, but what do the line colors mean?

The supplemental materials may not have been the final version, as track changes were still included.

Review form: Reviewer 2

Recommendation

Major revision is needed (please make suggestions in comments)

Scientific importance: Is the manuscript an original and important contribution to its field?

Good

General interest: Is the paper of sufficient general interest?

Good

Quality of the paper: Is the overall quality of the paper suitable?

Good

Is the length of the paper justified?

Yes

Should the paper be seen by a specialist statistical reviewer?

No

Do you have any concerns about statistical analyses in this paper? If so, please specify them explicitly in your report.

Yes

It is a condition of publication that authors make their supporting data, code and materials available - either as supplementary material or hosted in an external repository. Please rate, if applicable, the supporting data on the following criteria.

Is it accessible?

Yes

Is it clear?

Yes

Is it adequate?

Yes

Do you have any ethical concerns with this paper?

No

Comments to the Author

Review of Evans et al. "A natural catastrophic turnover event: individual sociality matters despite community resilience in wild house mice"

I very much enjoyed reading this manuscript. The study system itself is excellent, and provides some really unique data that can enable various insights which would usually be very difficult to study. The work itself is well written and carried out in a sensible manner, and looks at an interesting incidence of a major predation event. My main comments primarily revolve around the methods and the subsequent results (see Section 1), where I found that I was lacking some details to allow me to fully interpret the work, and also where I had some suggestions about how to potentially improve the analysis, and some pointers about which conclusions can confidently be drawn (given the set-up). I then also provide some suggestions in regards to text changes, first in relation to some potential over-statements (section 2a), and then some suggested areas that would benefit from some rewording (section 2b). Overall, I found the manuscript to be of much interest and I would very much enjoy reviewing the revised manuscript.

(Section 1) Methods & Results

Line 108-109: There needs to be more information on how these networks were made, is it proportion of seconds together, minutes, unique visits? Currently its hard to know what these networks signify.

Line 119: Why weren't multiple 'pre-attack' networks used for all of the analysis, instead of just for the community analysis? That way you would be able to see the expected rate of change in dynamics of the network, and how stable it is under normal conditions too

Line 124: In case you aren't aware, igraph inversely weights the networks when calculating weighted betweenness, so the stronger bonds are actually classed as weaker (it views them as a distance, not a bond strength). So, I think you will probably want to recalculate betweenness either using a different package, or by using igraph and explaining how this edge weight issue was corrected for.

Line 128: Why were the randomisations only carried out for the 'post-networks'? I think it is important to do this for the pre-network too (as it still holds the same issues of non-independence, incomparable structures etc etc).

Line 134: I think you need to start this section off this a short (1 sentence or so) description of why you wanted to run this method and its relation to your hypotheses (it isn't currently very intuitive to pick that up on the first read through).

Line 147-151: It is interesting to fit the 3-way interaction. But, I believe it would also be necessary to report the results of what happens when you just fit a 2-way interaction (where time-step is just a regular fixed factor, rather than interaction).

Line 152: Why did you choose these particular metrics? Some more info is needed on this

Line 159: It doesn't really make sense to compare these model outputs to the data-stream randomisations, because then the variance in the response variable between your observed data and your null data will be incomparable. It makes more sense to use node permutations for this.

Line 161: It would be best to report the actual standard effect sizes and p values, and also the effect sizes and p values as calculated from the null model comparison too.

Line 210-213: These descriptions of changes in the modularity value are interesting as descriptions, but they aren't tests of hypotheses. For instance, the modularity after the attack has to be either higher or lower than the modularity before the attack, but this doesn't mean that conclusions can be drawn on it. The extent (and significance) of this change in modularity is totally unknown. So, it is odd to then base conclusions (e.g. in the discussion: "The changes in size or density of connections within a cluster seemed unrelated to the loss of individuals within a group. However, we observed both a reduction in the modularity of the social structure and an increase in between community movements after the event. Thus, while the social clusters did not collapse they became less well defined and more permeable over the entire post-attack observation range. The implications of such stable group structure are that overall network structure may be far more resilient than assumed.") on these simple descriptions that can go either way.

This general point is true for all comparisons that these results make based purely just on the difference between two numbers. To make these kinds of conclusions we'd need many replications of this set-up and these events, so that we could actually say whether the difference is significant or not (as at least some difference is always expected).

So, the manuscript really needs to be careful to not draw any conclusions on purely descriptive values, and to clearly remind the reader that the study is based on just one event with no control.

(Section 2) Text changes:

(2a) Firstly, there are a few occasions where some statements are stated with undue certainty, I suggest rewording the following to be more exploratory-type statements:

Line 15: Change "as individuals lose connections" to "as individuals may lose connections" (as it's not certain that they definitely will)

Line 17: Change "Given that group membership and network position influence individual fitness" to "As group membership and network position may influence individual fitness" (because there is little/no definitive causal evidence that network position does influence fitness)

Line 18: Change "indirect effects will affect how individuals and populations recover from catastrophic events" to "indirect effects may affect how individuals and populations recover from catastrophic events" (due to above two reasons)

Line 25: Change "This indicates that social preferences will drive how individuals adjust their social behaviour after catastrophic turnover events" to "This indicates that social preferences can drive how individuals adjust their social behaviour after catastrophic turnover events" (as 'will' means it is general across systems)

(2b) There are some areas of the text that are unclear, would benefit from further explanation, or contain typos, I recommend these are changed as suggested below:

Line 21: I don't think there is any need for "Utilising dynamic community detection we investigate how these differences influence overall network structure" in the abstract as it's not really very informative, would be better to focus on a finding.

Line 24: It is difficult to know exactly what 'structural' is referring to in this sentence – I suggest rewording

Line 27-28: The final sentence of the abstract, and the term 'modularity', comes a bit out-of-the-blue here. I think someone reading this abstract for the first time would be confused by it (or not take much away from it).

Line 34: Change “potential dramatically” to “potential to dramatically”

Line 34: Change “impact a groups’ social structure” to either “impact a group’s social structure” OR “impact group social structure”

Line 48-49: This sentence could benefit from some more explanation (perhaps even another sentence added onto it explaining the definition of such effects).

Line 71: I think it is probably necessary to mention that these ‘natural experiments’ have the disadvantage (compared to actual experiments) that there is then just a sample size of one, and also that there is no control treatment. I think this needs to be clear early on in this MS.

Line 73-78: The whole section of text that reads

“In particular we focus on the indirect effects that the loss of connections has on remaining individuals, and how this varies based on the connections lost and inherent heterogeneity in individual sociality prior to the event. Additionally, using dynamic community analysis we describe the changes in groups in the aftermath of the event, allowing us to examine changes over time at both the individual and community level”

Doesn’t really add much at this point in the manuscript. I doubt that most readers will know what this means or why it is in this section of the introduction.

Line 105: Change “as attack took” to “as the attack took”

Line 108: Change “simple ratio indexes” to “the simple ratio index” (and cite the source of this index)

Line 128: Saying “hypothesis testing purposes” is a bit to vague.

Line 234-236: This is too vague too, it needs to say how/in what way they altered their interactions

Line 269: Why would it be expected that individuals would start interacting randomly after perturbation?

I hope the authors find these comments useful. I congratulate them again on an interesting study, and would be happy to review a revised version of the manuscript.

Best wishes

Josh Firth (please note, I sign all my reviews).

Decision letter (RSPB-2019-2880.R0)

07-Feb-2020

Dear Julian:

Your manuscript has now been peer reviewed and the reviews have been assessed by an Associate Editor. The reviewers’ comments (not including confidential comments to the Editor) and the comments from the Associate Editor are included at the end of this email for your reference. As you will see, the reviewers and the Editors have raised some concerns with your manuscript and we would like to invite you to revise your manuscript to address them.

We do not allow multiple rounds of revision so we urge you to make every effort to fully address all of the comments at this stage. If deemed necessary by the Associate Editor, your manuscript

will be sent back to one or more of the original reviewers for assessment. If the original reviewers are not available we may invite new reviewers. Please note that we cannot guarantee eventual acceptance of your manuscript at this stage.

Research ethics:

Use of animals and field studies:

Please submit a copy of your revised paper within three weeks. If we do not hear from you within this time your manuscript will be rejected. If you are unable to meet this deadline please let us know as soon as possible, as we may be able to grant a short extension.

Best wishes,
Sasha

Dr Sasha Dall
mailto:proceedingsb@royalsociety.org

Associate Editor
Comments to Author:

I have now received reviews from two experts. Both reviewers were positive about the topic and the potential for this study to be a substantial contribution to the field. However, both reviewers outlined specific concerns regarding other aspects of the environment, the analyses, and the interpretations. From my own reading of the manuscript, I am in agreement with the reviewer comments and concerns.

Reviewer 1 highlights that additional information about the environment, particularly aspects related to distribution of resources, should be included and discussed. I too had naively assumed that the barn was a single open room. It would also be helpful for some basic mice background to be included, such as how many mice occupy nests, whether these are mixed sex/age, how these groups share/use resources, etc... as well as why 5days is an appropriate time period from which to estimate associations and networks.

Reviewer 2 has many excellent suggestions regarding the analyses and their interpretation. I was also curious about how sex/age might have affected the relationships among individuals, particularly since it is unclear whether there are sex-biased association patterns and whether predation affected both sexes equally. Can sex be examined in these analyses (or at least excluded as a major factor driving patterns)? Inclusion of the numbers of each sex that were predated would also be informative.

I would also encourage the authors to include brief explanations for each of the network metrics (e.g. degree, weighted degree, betweenness centrality, edge density/strength) where appropriate (but before the discussion) and describe what they mean in biological terms for readers not familiar with network analysis. Additionally, as pointed out in the reviews, figure headings should be clarified so that all parts, including colors, are interpretable and the figures can stand on their own.

Overall, I think this unique study would appeal to a broad audience and I encourage the authors to pay careful attention to the helpful comments so thoughtfully provided by the reviewers.

Minor comments

1. Line 34: "...has the potential TO dramatically..."
2. Line 48: This sentence highlights "indirect effects" of losing social connections. Are the following few sentences examples of these indirect effects? It is not clear.
3. Line 63: replace "larger" with "...over a LONGER period of time..."
4. Figure 1 - sample sizes should be included for the red/purple/blue mice
5. Line 190: Table 1 is missing.
6. Line 190: remove period after "...before the event"
7. Line 270: remove extra comma after citation.

Katie McGhee
Associate Editor

Reviewer(s)' Comments to Author:

Referee: 1

Comments to the Author(s)

This study leverages a catastrophic predation event in a well-studied wild population of house mice to explore the consequences of sudden, massive population decline on social network structure. Fortunately, the social structure of the population had been comprehensively monitored using a continuous RFID data logging system as part of a long-term study prior to the predation event. Since the monitoring could continue with the surviving mice, the researchers could analyze in detail the stability of the social structure despite the loss of about 40% of the population. They also find that an individual's pattern of sociality prior to the predation event affects how their patterns of connectivity change after the event. This is certainly an interesting case study for linking demographic events to social structure in natural populations.

One inevitable limitation of this study, as with any study relying solely on RFID systems, is that the assessment of social structure is necessarily tied to specific locations seeded with key resources. Thus, the stability of the social network may be purely driven by the stability of an individual's preference for the nest boxes that it knows. This would be meaningful in its own way, so it doesn't necessarily take away from the study. However, the spatial aspect of network stability should be acknowledged throughout this paper. In the Discussion, it may be worth also considering how response to catastrophic mortality events in location-based social networks like this one may differ from other systems in which social associations can occur over broader spatial areas.

More minor comments:

Lines 45-49: The two sentences here sound redundant with each other. Perhaps there needs to be more explanation of what "indirect effects" are, and how they are distinct from loss of connections due to death.

Line 94: The spatial structure of the barn and nest box array should be described in the paper. If there is no room, it could be presented as a supplement. Without this detail, it makes the overall patterns more challenging to interpret. For example, it is stated here that the mice are free to move and disperse. Upon reading the study that describes the methodology in full (König et al. 2015), it seems that the setup is slightly more complex. For example, the barn is segmented into 4 parts, partitioned by walls that have holes that mice can move through. While this may not limit dispersal per se, it likely restricts movement patterns significantly, which would likely

change how individuals use nest boxes. I'm not saying that this is necessarily a major flaw in the study – any natural system will also have structures that affect movement patterns, and thus social structure. My main point is that this spatial arrangement should be described within this paper in order to make it easier for the readers to interpret the results.

Line 190: I did not see Table 1 in the manuscript.

Line 190-194; Lines 200-202: It took me a while to get that these sentences are referring to the changes in the slope or intercept in panels in Figure 4 going left to right on each row. I think there needs to be much more explanation either in the main text or in the figure captions.

Line 221: “Unsurprisingly” □ “As predicted”?

Figure 4: I could not really interpret this figure because the color scales were not explained. The color gradient on the right seems to correspond to the color of points, but what do the line colors mean?

The supplemental materials may not have been the final version, as track changes were still included.

Referee: 2

Comments to the Author(s)

Review of Evans et al. “A natural catastrophic turnover event: individual sociality matters despite community resilience in wild house mice”

I very much enjoyed reading this manuscript. The study system itself is excellent, and provides some really unique data that can enable various insights which would usually be very difficult to study. The work itself is well written and carried out in a sensible manner, and looks at an interesting incidence of a major predation event. My main comments primarily revolve around the methods and the subsequent results (see Section 1), where I found that I was lacking some details to allow me to fully interpret the work, and also where I had some suggestions about how to potentially improve the analysis, and some pointers about which conclusions can confidently be drawn (given the set-up). I then also provide some suggestions in regards to text changes, first in relation to some potential over-statements (section 2a), and then some suggested areas that would benefit from some rewording (section 2b). Overall, I found the manuscript to be of much interest and I would very much enjoy reviewing the revised manuscript.

(Section 1) Methods & Results

Line 108-109: There needs to be more information on how these networks were made, is it proportion of seconds together, minutes, unique visits? Currently its hard to know what these networks signify.

Line 119: Why weren't multiple 'pre-attack' networks used for all of the analysis, instead of just for the community analysis? That way you would be able to see the expected rate of change in dynamics of the network, and how stable it is under normal conditions too

Line 124: In case you aren't aware, igraph inversely weights the networks when calculating weighted betweenness, so the stronger bonds are actually classed as weaker (it views them as a distance, not a bond strength). So, I think you will probably want to recalculate betweenness either using a different package, or by using igraph and explaining how this edge weight issue was corrected for.

Line 128: Why were the randomisations only carried out for the 'post-networks'? I think it is

important to do this for the pre-network too (as it still holds the same issues of non-independence, incomparable structures etc etc).

Line 134: I think you need to start this section off with a short (1 sentence or so) description of why you wanted to run this method and its relation to your hypotheses (it isn't currently very intuitive to pick that up on the first read through).

Line 147-151: It is interesting to fit the 3-way interaction. But, I believe it would also be necessary to report the results of what happens when you just fit a 2-way interaction (where time-step is just a regular fixed factor, rather than interaction).

Line 152: Why did you choose these particular metrics? Some more info is needed on this

Line 159: It doesn't really make sense to compare these model outputs to the data-stream randomisations, because then the variance in the response variable between your observed data and your null data will be incomparable. It makes more sense to use node permutations for this.

Line 161: It would be best to report the actual standard effect sizes and p values, and also the effect sizes and p values as calculated from the null model comparison too.

Line 210-213: These descriptions of changes in the modularity value are interesting as descriptions, but they aren't tests of hypotheses. For instance, the modularity after the attack has to be either higher or lower than the modularity before the attack, but this doesn't mean that conclusions can be drawn on it. The extent (and significance) of this change in modularity is totally unknown. So, it is odd to then base conclusions (e.g. in the discussion: "The changes in size or density of connections within a cluster seemed unrelated to the loss of individuals within a group. However, we observed both a reduction in the modularity of the social structure and an increase in between community movements after the event. Thus, while the social clusters did not collapse they became less well defined and more permeable over the entire post-attack observation range. The implications of such stable group structure are that overall network structure may be far more resilient than assumed.") on these simple descriptions that can go either way.

This general point is true for all comparisons that these results make based purely just on the difference between two numbers. To make these kinds of conclusions we'd need many replications of this set-up and these events, so that we could actually say whether the difference is significant or not (as at least some difference is always expected).

So, the manuscript really needs to be careful to not draw any conclusions on purely descriptive values, and to clearly remind the reader that the study is based on just one event with no control.

(Section 2) Text changes:

(2a) Firstly, there are a few occasions where some statements are stated with undue certainty, I suggest rewording the following to be more exploratory-type statements:

Line 15: Change "as individuals lose connections" to "as individuals may lose connections" (as it's not certain that they definitely will)

Line 17: Change "Given that group membership and network position influence individual fitness" to "As group membership and network position may influence individual fitness" (because there is little/no definitive causal evidence that network position does influence fitness)

Line 18: Change "indirect effects will affect how individuals and populations recover from catastrophic events" to "indirect effects may affect how individuals and populations recover from catastrophic events" (due to above two reasons)

Line 25: Change “This indicates that social preferences will drive how individuals adjust their social behaviour after catastrophic turnover events” to “This indicates that social preferences can drive how individuals adjust their social behaviour after catastrophic turnover events” (as ‘will’ means it is general across systems)

(2b) There are some areas of the text that are unclear, would benefit from further explanation, or contain typos, I recommend these are changed as suggested below:

Line 21: I don’t think there is any need for “Utilising dynamic community detection we investigate how these differences influence overall network structure” in the abstract as its not really very informative, would be better to focus on a finding.

Line 24: It is difficult to know exactly what ‘structural’ is referring to in this sentence – I suggest rewording

Line 27-28: The final sentence of the abstract, and the term ‘modularity’, comes a bit out-of-the-blue here. I think someone reading this abstract for the first time would be confused by it (or not take much away from it).

Line 34: Change “potential dramatically” to “potential to dramatically”

Line 34: Change “impact a groups’ social structure” to either “impact a group’s social structure” OR “impact group social structure”

Line 48-49: This sentence could benefit from some more explanation (perhaps even another sentence added onto it explaining the definition of such effects).

Line 71: I think it is probably necessary to mention that these ‘natural experiments’ have the disadvantage (compared to actual experiments) that there is then just a sample size of one, and also that there is no control treatment. I think this needs to be clear early on in this MS.

Line 73-78: The whole section of text that reads

“In particular we focus on the indirect effects that the loss of connections has on remaining individuals, and how this varies based on the connections lost and inherent heterogeneity in individual sociality prior to the event. Additionally, using dynamic community analysis we describe the changes in groups in the aftermath of the event, allowing us to examine changes over time at both the individual and community level”

Doesn’t really add much at this point in the manuscript. I doubt that most readers will know what this means or why it is in this section of the introduction.

Line 105: Change “as attack took” to “as the attack took”

Line 108: Change “simple ratio indexes” to “the simple ratio index” (and cite the source of this index)

Line 128: Saying “hypothesis testing purposes” is a bit too vague.

Line 234-236: This is too vague too, it needs to say how/in what way they altered their interactions

Line 269: Why would it be expected that individuals would start interacting randomly after perturbation?

I hope the authors find these comments useful. I congratulate them again on an interesting study, and would be happy to review a revised version of the manuscript.

Best wishes

Josh Firth (please note, I sign all my reviews).

Author's Response to Decision Letter for (RSPB-2019-2880.R0)

See Appendix A.

Decision letter (RSPB-2019-2880.R1)

03-Apr-2020

Dear Julian

I am pleased to inform you that your Review manuscript RSPB-2019-2880.R1 entitled "A natural catastrophic turnover event: individual sociality matters despite community resilience in wild house mice" has been accepted for publication in Proceedings B.

The referee(s) do not recommend any further changes. Therefore, please proof-read your manuscript carefully and upload your final files for publication. Because the schedule for publication is very tight, it is a condition of publication that you submit the revised version of your manuscript within 7 days. If you do not think you will be able to meet this date please let me know immediately.

To upload your manuscript, log into <http://mc.manuscriptcentral.com/prsb> and enter your Author Centre, where you will find your manuscript title listed under "Manuscripts with Decisions." Under "Actions," click on "Create a Revision." Your manuscript number has been appended to denote a revision.

You will be unable to make your revisions on the originally submitted version of the manuscript. Instead, upload a new version through your Author Centre.

1) A text file of the manuscript (doc, txt, rtf or tex), including the references, tables (including captions) and figure captions. Please remove any tracked changes from the text before submission. PDF files are not an accepted format for the "Main Document".

2) A separate electronic file of each figure (tiff, EPS or print-quality PDF preferred). The format should be produced directly from original creation package, or original software format. Please note that PowerPoint files are not accepted.

3) Electronic supplementary material: this should be contained in a separate file from the main text and the file name should contain the author's name and journal name, e.g. `authorname_procb_ESM_figures.pdf`

All supplementary materials accompanying an accepted article will be treated as in their final form. They will be published alongside the paper on the journal website and posted on the online figshare repository. Files on figshare will be made available approximately one week before the

accompanying article so that the supplementary material can be attributed a unique DOI. Please see: <https://royalsociety.org/journals/authors/author-guidelines/>

4) Data-Sharing and data citation

It is a condition of publication that data supporting your paper are made available. Data should be made available either in the electronic supplementary material or through an appropriate repository. Details of how to access data should be included in your paper. Please see <https://royalsociety.org/journals/ethics-policies/data-sharing-mining/> for more details.

<http://datadryad.org/submit?journalID=RSPB&manu=RSPB-2019-2880.R1> which will take you to your unique entry in the Dryad repository.

Once again, thank you for submitting your manuscript to Proceedings B and I look forward to receiving your final version. If you have any questions at all, please do not hesitate to get in touch.

Sincerely,
Sasha

Dr Sasha Dall
Editor, Proceedings B
<mailto:proceedingsb@royalsociety.org>

Associate Editor
Comments to Author:

Thank you for your careful attention to the reviewer comments. I feel that the revised version of the manuscript will be of broad interest and an excellent contribution to Proc B!

Minor comments:

1. Line 122: "...were chosen SO as to..."
2. Line 132: "...betweenNess..."
3. Line 145: "...expecting it TO decrease..."
4. Line 146: Divide into two separate sentences: "...when measuring betweenness. We accounted for this..."
5. Line 164 " ...to measure changes IN the size of ..."
6. Line 234: " ...similar TO what they had..."
7. Line 307: two sentences? "...a new set of individuals. INDIVIDUALS might BE particularly driven to..."
8. Line 333: " ...expected to see an increase in AN individual's betweenness..."
9. Line 386: add in comma " ...following breeding season, will help..."

Katie McGhee
Associate Editor

Decision letter (RSPB-2019-2880.R2)

11-Apr-2020

Dear Julian

I am pleased to inform you that your manuscript entitled "A natural catastrophic turnover event: individual sociality matters despite community resilience in wild house mice" has been accepted for publication in Proceedings B.

Open Access
You are invited to opt for Open Access, making your article freely available to all as soon as it is ready for publication under a CCBY licence. Our article processing charge for Open Access is £1700. Corresponding authors from member institutions (<http://royalsocietypublishing.org/site/librarians/allmembers.xhtml>) receive a 25% discount to these charges. For more information please visit <http://royalsocietypublishing.org/open-access>.

Paper charges

Sincerely,
Sasha

Dr Sasha Dall
Editor, Proceedings B
<mailto:proceedingsb@royalsociety.org>

Associate Editor:

Board Member

Comments to Author:

Thank you for your attention to all of my comments and those provided by the reviewers.

Congratulations on an excellent contribution! A bit of good news in a crazy time!

Best,

Katie McGhee

Associate Editor

Appendix A

Comments to Author:

I have now received reviews from two experts. Both reviewers were positive about the topic and the potential for this study to be a substantial contribution to the field. However, both reviewers outlined specific concerns regarding other aspects of the environment, the analyses, and the interpretations. From my own reading of the manuscript, I am in agreement with the reviewer comments and concerns.

Reviewer 1 highlights that additional information about the environment, particularly aspects related to distribution of resources, should be included and discussed. I too had naively assumed that the barn was a single open room. It would also be helpful for some basic mice background to be included, such as how many mice occupy nests, whether these are mixed sex/age, how these groups share/use resources, etc... as well as why 5days is an appropriate time period from which to estimate associations and networks.

We have now added a map and a photo and some additional details about the study area throughout. We also justify our use of a 5 day time step.

Reviewer 2 has many excellent suggestions regarding the analyses and their interpretation. I was also curious about how sex/age might have affected the relationships among individuals, particularly since it is unclear whether there are sex-biased association patterns and whether predation affected both sexes equally. Can sex be examined in these analyses (or at least excluded as a major factor driving patterns)? Inclusion of the numbers of each sex that were predated would also be informative.

We agree that this would be informative and is certainly something we are interested in. We originally avoided including sex in the presented analysis as we felt there was already quite a lot going, and we felt that any differences between sexes would be included in our social phenotype measure (males tend to be less social than females etc.).

We now report individual sexes, as suggested. We also present an analysis where we substitute our social phenotype for sex. Though most effects were weak, there were some suggested differences in how males change their weighted degree.

Between this and reviewer 2's comments, we have a large number of models. Due to this, we have decided to remove the extra analyses originally included where we repeated analysis using only individuals who were found dead, rather than all missing individuals, as these models did not really differ from the main analyses presented in the paper.

I would also encourage the authors to include brief explanations for each of the network metrics (e.g. degree, weighted degree, betweenness centrality, edge density/strength) where appropriate (but before the discussion) and describe what they mean in biological terms for readers not familiar with network analysis. Additionally, as pointed out in the reviews, figure headings should be clarified so that all parts, including colors, are interpretable and the figures can stand on their own.

We now give an overview of these metrics and the sort of changes we expected and what they would signify biologically in the methods. We have also substantially changed figure 4, including both a pre-event timestep as suggested by reviewer 2 and additional information in the figure to make it clearer. In order to make room for these changes, we removed figure 4. c, which showed no clear patterns or significant effects.

Overall, I think this unique study would appeal to a broad audience and I encourage the authors to pay careful attention to the helpful comments so thoughtfully provided by the reviewers.

We thank the editor for their positive and very helpful comments!

Minor comments

1. Line 34: "...has the potential TO dramatically..."

We have corrected this,

2. Line 48: This sentence highlights "indirect effects" of losing social connections. Are the following few sentences examples of these indirect effects? It is not clear.

We have reworded and expanded this to make it clearer that the following sentences are examples of this and that individuals are affected by subsequent alterations in social structure (Lines 44 – 52)

3. Line 63: replace "larger" with "...over a LONGER period of time..."

We have made this correction

4. Figure 1 – sample sizes should be included for the red/purple/blue mice

This information has now been added

5. Line 190: Table 1 is missing.

This was meant to reference a supplementary table, this has now been corrected.

6. Line 190: remove period after "...before the event"

We have made this correction

7. Line 270: remove extra comma after citation.

We have made this correction

Katie McGhee
Associate Editor

Reviewer(s)' Comments to Author:

Referee: 1

Comments to the Author(s)

This study leverages a catastrophic predation event in a well-studied wild population of house mice to explore the consequences of sudden, massive population decline on social network structure. Fortunately, the social structure of the population had been comprehensively monitored using a continuous RFID data logging system as part of a long-term study prior to the predation event. Since the monitoring could continue with the surviving mice, the researchers could analyze in detail the stability of the social structure despite the loss of about 40% of the population. They also find that an individual's pattern of sociality prior to the predation event affects how their patterns of connectivity change after the event. This is certainly an interesting case study for linking demographic events to social structure in natural populations.

One inevitable limitation of this study, as with any study relying solely on RFID systems, is that the assessment of social structure is necessarily tied to specific locations seeded with key resources. Thus, the stability of the social network may be purely driven by the stability of an individual's preference for the nest boxes that it knows. This would be meaningful in its own way, so it doesn't necessarily take away from the study. However, the spatial aspect of network stability should be acknowledged throughout this paper. In the Discussion, it may be worth also considering how response to catastrophic mortality events in location-based social networks like this one may differ from other systems in which social associations can occur over broader spatial areas.

We thank the Associate Editor and all reviewers for their positive comments. We fully agree that spatial structure will play a large part in the stability of social structure. We have added some extra discussion of this throughout as well as incorporating the suggestion that we speculate about how location based social networks may differ from networks where interactions can happen more freely (Lines 357-362).

More minor comments:

Lines 45-49: The two sentences here sound redundant with each other. Perhaps there needs to be more explanation of what "indirect effects" are, and how they are distinct from loss of connections due to death.

We made clearer that indirect effects are the alteration of social connections with other survivors as opposed to the loss of connections with those who have left/died (Lines 44 – 52)

Line 94: The spatial structure of the barn and nest box array should be described in the paper. If there is no room, it could be presented as a supplement. Without this detail, it makes the overall patterns more challenging to interpret. For example, it is stated here that the mice are free to move and disperse. Upon reading the study that describes the methodology in full (König et al. 2015), it seems that the setup is slightly more complex. For example, the barn is segmented into 4 parts, partitioned by walls that have holes that mice can move through. While this may not limit dispersal per se, it likely restricts movement patterns significantly, which would likely change how individuals use nest boxes. I'm not saying that this is necessarily a major flaw in the study—any natural system will also have structures that

affect movement patterns, and thus social structure. My main point is that this spatial arrangement should be described within this paper in order to make it easier for the readers to interpret the results.

The reviewer is correct that spatial structure will have a definite effect on the patterns observed and that the reader should ideally have all the details about this within the paper. We have added details about the barn layout (lines 86-91), and a map/photo in the SI that should hopefully make the structure clearer to readers. We have also added details (Line 271 – 277) and discussion (Lines 304-310) about whether the movements between social group in the barn are between groups in different sections.

Line 190: I did not see Table 1 in the manuscript.

This was supposed to be a reference to a Supplementary table, we have corrected this.

Line 190-194; Lines 200-202: It took me a while to get that these sentences are referring to the changes in the slope or intercept in panels in Figure 4 going left to right on each row. I think there needs to be much more explanation either in the main text or in the figure captions.

We have now modified the text throughout this section and altered the figure to be clearer.

Line 221: “Unsurprisingly” \diamond “As predicted”?

We have incorporated this suggestion.

Figure 4: I could not really interpret this figure because the color scales were not explained. The color gradient on the right seems to correspond to the color of points, but what do the line colors mean?

We have now stated that the point and line colours are on the same colour scale, and thus the lines represent min, mean and max levels of pre-event sociality. We have also reworked the figure to be clearer.

The supplemental materials may not have been the final version, as track changes were still included.

Referee: 2

Comments to the Author(s)

Review of Evans et al. “A natural catastrophic turnover event: individual sociality matters despite community resilience in wild house mice”

I very much enjoyed reading this manuscript. The study system itself is excellent, and provides some really unique data that can enable various insights which would usually be very difficult to study. The work itself is well written and carried out in a sensible manner, and looks at an interesting incidence of a major predation event. My main comments primarily revolve around the methods and the subsequent results (see Section 1), where I found that I was lacking some details to allow me to fully interpret the work, and also where I had some suggestions about how to potentially improve the analysis, and some pointers about which conclusions can confidently be drawn (given the set-up). I then also provide some suggestions in regards to text changes, first in relation to some potential over-statements (section 2a), and then some suggested areas that would benefit from some rewording (section 2b). Overall, I found the manuscript to be of much interest and I would very much enjoy reviewing the revised manuscript.

(Section 1) Methods & Results

Line 108-109: There needs to be more information on how these networks were made, is it proportion of seconds together, minutes, unique visits? Currently its hard to know what these networks signify.

We now state that the times were accurate to the second, and that it is the proportion of time these two individuals spent in the same nest boxes out of their total nest-box use (Line 115).

Line 119: Why weren't multiple 'pre-attack' networks used for all of the analysis, instead of just for the community analysis? That way you would be able to see the expected rate of change in dynamics of the network, and how stable it is under normal conditions too.

This is something we have also come round to since submitting this paper. Including the timestep immediately prior to the attack as the intercept in our model makes a lot of sense in terms of being able to quantify the actual effect of the attack rather than just the recovery (previously the first timestep immediately after the attack was the intercept). The models have been rerun using all timesteps and methods, results and figures altered accordingly.

Line 124: In case you aren't aware, igraph inversely weights the networks when calculating weighted betweenness, so the stronger bonds are actually classed as weaker (it views them as a distance, not a bond strength). So, I think you will probably want to recalculate betweenness either using a different package, or by using igraph and explaining how this edge weight issue was corrected for.

We were aware of this issue and corrected for it. We now state this on line 144.

Line 128: Why were the randomisations only carried out for the 'post-networks'? I think it is important to do this for the pre-network too (as it still holds the same issues of non-independence, incomparable structures etc etc).

At the time we felt that in terms of what we were testing, this would ask the question "Did the responses of individuals (based on pre-attack weighted degree, proportion individuals lost etc.) differ from a purely random rewiring". We might expect this if individuals were just running about and panicking, joining any available group for the dilution effect or just moving to a different area of the barn because they no longer perceive theirs as safe. Thus, we only randomised the post networks (now also the other pre-event timestep). We thought that if we also randomised the network we used to

compare all other networks to and generate pre-event sociality we'd be randomising both the response and explanatory variables and thus be comparing how a random network differed from another random network, based on another random network. We didn't think this would be very biologically meaningful, other than showing whether our data in general differed from random behaviours.

Line 134: I think you need to start this section off with a short (1 sentence or so) description of why you wanted to run this method and its relation to your hypotheses (it isn't currently very intuitive to pick that up on the first read through).

We have now added a sentence explaining our motives for using this analysis line 160.

Line 147-151: It is interesting to fit the 3-way interaction. But, I believe it would also be necessary to report the results of what happens when you just fit a 2-way interaction (where time-step is just a regular fixed factor, rather than interaction).

We have now added these models, they definitely help understand the patterns better I think.

Line 152: Why did you choose these particular metrics? Some more info is needed on this

We have now added extra details about our reasoning/predictions behind these metrics (lines 131 - 143).

Line 159: It doesn't really make sense to compare these model outputs to the data-stream randomisations, because then the variance in the response variable between your observed data and your null data will be incomparable. It makes more sense to use node permutations for this.

We have redone the randomisations with node permutations and altered methods and results accordingly.

Line 161: It would be best to report the actual standard effect sizes and p values, and also the effect sizes and p values as calculated from the null model comparison too.

We now include the mean and SD random effect sizes in the tables to give an idea of the distribution of the randomised results.

Line 210-213: These descriptions of changes in the modularity value are interesting as descriptions, but they aren't tests of hypotheses. For instance, the modularity after the attack has to be either higher or lower than the modularity before the attack, but this doesn't mean that conclusions can be drawn on it. The extent (and significance) of this change in modularity is totally unknown. So, it is odd to then base conclusions (e.g. in the discussion: "The changes in size or density of connections within a cluster seemed unrelated to the loss of individuals within a group. However, we observed both a reduction in the modularity of the social structure and an increase in between community movements after the event. Thus, while the social clusters did not collapse they became less well defined and more permeable over the entire post-attack observation range. The implications of such stable group

structure are that overall network structure may be far more resilient than assumed.”) on these simple descriptions that can go either way.

This general point is true for all comparisons that these results make based purely just on the difference between two numbers. To make these kinds of conclusions we’d need many replications of this set-up and these events, so that we could actually say whether the difference is significant or not (as at least some difference is always expected).

So, the manuscript really needs to be careful to not draw any conclusions on purely descriptive values, and to clearly remind the reader that the study is based on just one event with no control.

Agreed, this was not our intention as we have no formal analysis on this! We have altered language accordingly in the discussion. We have weakened the language in the above paragraph in which we link the models of community change with the descriptive measures above, and are more cautious with the conclusions drawn.

(Section 2) Text changes:

(2a) Firstly, there are a few occasions where some statements are stated with undue certainty, I suggest rewording the following to be more exploratory-type statements:

Line 15: Change “as individuals lose connections” to “as individuals may lose connections”
(as its not certain that they definitely will)

We have implemented this change.

Line 17: Change “Given that group membership and network position influence individual fitness” to “As group membership and network position may influence individual fitness”
(because there is little/no definitive causal evidence that network position does influence fitness)

We have implemented this change.

Line 18: Change “indirect effects will affect how individuals and populations recover from catastrophic events” to “indirect effects may affect how individuals and populations recover from catastrophic events”
(due to above two reasons)

We have implemented this change.

Line 25: Change “This indicates that social preferences will drive how individuals adjust their social behaviour after catastrophic turnover events” to “This indicates that social preferences can drive how individuals adjust their social behaviour after catastrophic turnover events”
(as ‘will’ means it is general across systems)

We have implemented this change.

(2b) There are some areas of the text that are unclear, would benefit from further explanation, or contain typos, I recommend these are changed as suggested below:

Line 21: I don't think there is any need for "Utilising dynamic community detection we investigate how these differences influence overall network structure" in the abstract as its not really very informative, would be better to focus on a finding.

We have removed this as suggested.

Line 24: It is difficult to know exactly what 'structural' is referring to in this sentence – I suggest rewording

We have reworded to make it clearer that we are discussing the social groups.

Line 27-28: The final sentence of the abstract, and the term 'modularity', comes a bit out-of-the-blue here. I think someone reading this abstract for the first time would be confused by it (or not take much away from it).

We have removed this sentence.

Line 34: Change "potential dramatically" to "potential to dramatically"

We have implemented this change.

Line 34: Change "impact a groups' social structure" to either "impact a group's social structure" OR "impact group social structure"

We have implemented this change and used "impact group social structure".

Line 48-49: This sentence could benefit from some more explanation (perhaps even another sentence added onto it explaining the definition of such effects).

We have altered this section to make the difference in these effects and how they might effect individuals clearer (lines 44-52)

Line 71: I think it is probably necessary to mention that these 'natural experiments' have the disadvantage (compared to actual experiments) that there is then just a sample size of one, and also that there is no control treatment. I think this needs to be clear early on in this MS.

We now mention this and the additional benefit that removal experiments can target specifically target particular individuals on line 66-67.

Line 73-78: The whole section of text that reads

"In particular we focus on the indirect effects that the loss of connections has on remaining individuals,

and how this varies based on the connections lost and inherent heterogeneity in individual sociality prior to the event. Additionally, using dynamic community analysis we describe the changes in groups in the aftermath of the event, allowing us to examine changes over time at both the individual and community level”

Doesn't really add much at this point in the manuscript. I doubt that most readers will know what this means or why it is in this section of the introduction.

We have removed this section

Line 105: Change “as attack took” to “as the attack took”

We have implemented this change.

Line 108: Change “simple ratio indexes” to “the simple ratio index” (and cite the source of this index)

We have made this change and added the appropriate citation.

Line 128: Saying “hypothesis testing purposes” is a bit to vague.

This section (line 149-158) has been rewritten, giving examples of how individuals might behave more randomly.

Line 234-236: This is too vague too, it needs to say how/in what way they altered their interactions

We now specifically state the changes (line 290-292)

Line 269: Why would it be expected that individuals would start interacting randomly after perturbation?

We have expanded on potential reasons why we might expect more random interactions/encounters such as increased movement due to stress, reduced choosiness in associates or reduced space use due the perceived risk in that area.

I hope the authors find these comments useful. I congratulate them again on an interesting study, and would be happy to review a revised version of the manuscript.

We thank the reviewer for their extremely helpful and constructive comments and suggestions.